# Band Phosphorus and Sulfur Fertilization as Drivers of Efficient Management of Nitrogen of Maize (*Zea mays* L.)

**DOI:** 10.3390/plants11131660

**Published:** 2022-06-23

**Authors:** Przemysław Barłóg, Remigiusz Łukowiak, Lukáš Hlisnikovský

**Affiliations:** 1Department of Agricultural Chemistry and Environmental Biogeochemistry, Poznan University of Life Sciences, Wojska Polskiego 71F, 60-625 Poznan, Poland; remigiusz.lukowiak@up.poznan.pl; 2Department of Nutrition Management, Crop Research Institute, Drnovská 507, Ruzyně, 161-01 Prague, Czech Republic; l.hlisnik@vurv.cz

**Keywords:** agronomic efficiency, nitrogen gap, nitrogen remobilization efficiency, partial factor productivity, plant nutrient diagnosis, starter fertilization

## Abstract

Increasing the efficiency of nitrogen use (NUE) from mineral fertilizers is one of the most important priorities of modern agriculture. The objectives of the present study were to assess the role of different nitrogen (N), phosphorus (P) and sulfur (S) rates on maize grain yield (GY), crop residue biomass, NUE indices, N concentration in plants during the growing season, N management indices and to select the most suitable set of NUE indicators. The following factors were tested: band application of di-ammonium phosphate and ammonium sulphate mixture (NPS fertilizer at rates 0, 8.7, 17.4, 26.2 kg ha^−1^ of P) and different total N rates (0, 60, 120, 180 kg ha^−1^ of N). In each year of the study, a clear trend of increased GY after NP(S) band application was observed. A particularly positive influence of that factor was confirmed at the lowest level of N fertilization. On average, the highest GY values were obtained for N2P3 and N3P1 treatments. The total N uptake and NUE indices also increased after the band application. In addition, a trend of improved N remobilization efficiency and the N contribution of remobilized N to grain as a result of band application of NP(S) was observed. Among various NUE indices, internal N utilization efficiency (IE) exhibited the strongest, yet negative, correlation with GY, whereas IE was a function of the N harvest index.

## 1. Introduction

Maize is one of the most important crops in the world. Among all cereals, it ranks second in terms of cultivated area (197 million hectares), just behind wheat (216 million hectares). Nevertheless, its global production between 2018 and 2020 was 1137 million tonnes, which was approximately 50% higher than the production of wheat [1]. According to the forecast, the global maize area in 2030 will expand even more (+5%), and the average yield, due to improving technology and cultivation practices, will increase by 10% [2]. In Poland, maize is also one of the dominant species. The area of maize grown for grain is about 1.0 million hectares, and maize for silage covers approximately 0.68 million hectares [3]. The yielding potentials of modern grain crop varieties are 11.6–12.8 t ha^−1^ [4]. In agricultural reality, however, they are lower (7.0–7.5 t ha^−1^) and constitute about 60% of the breeding potential. There are many reasons for this state of affairs, one of which is inadequate agro-technology, a low level of fertilization and an imbalance of minerals.

The main factor determining photosynthesis, dry matter distribution and water efficiency is the concentration of soil available nitrogen (N_min_) as well as the running N fertilization [5,6]. Unfortunately, the recovery of N from applied fertilizers (N_f_) ranged from 30 to 50% only [7]. The part of N_f_ that is not consumed by the currently grown crop undergoes numerous processes that result in its loss to neighboring ecosystems, including both water and air [8,9]. Crop genetic improvements along with agronomic attempts to take control of N management in agriculture comprise a set of different strategies, which, in fact, focus on the increase in nitrogen use efficiency (NUE) [10,11,12].

At the beginning of plant growth, standard broadcast fertilization does not always ensure proper plant nutrition because, depending on soil properties, part of the component introduced into the soil in the form of fertilizer will land in places that are beyond the range of crop roots [13]. An alternative fertilization method is to place the fertilizer in close proximity to the seeds. This type of fertilizer application, also referred to as fertilizer placement, band application or initial, significantly accelerates the growth of maize and improves the plants’ nutritional status at the beginning of the growing season [14]. This is due to both the increased concentration of minerals in close proximity of developing seedlings and the shaping of root architecture by N and P in particular [15,16,17]. In addition, this method of nitrogen application places the nutrient in a deeper, wetter soil layer, resulting in improved N uptake and limited N losses out to the environment [18]. Consequently, band placement not only prompts yield growth but also increases the values of NUE indices [17,19]. In general, fertilizer placement leads to an increase in maize yield in comparison to broadcast irrespective of the N fertilizer type [20]. Nevertheless, the use of ammonium phosphate proves to be the most productive of methods [21,22,23]. Despite numerous scientific findings on the beneficial impact of band application on maize yield and NUE, the N ratio still poses a problem and is little recognized, especially in conditions of high N demand. Apart from the many advantages of band fertilization, sprouting plants may become damaged, particularly when high rates of N fertilizers are used, where N comes in the form of NH_4_-N [24].

Another problem associated with improving the use of N from fertilizers is the correct balance of nutrients. In temperate climate conditions, phosphorus (P) is of particular importance in maize cultivation. The initial growth of maize is slow; the root system is poorly developed, which causes low uptake of nutrients, including nitrogen, and at the same time, periodic temperature drops in the spring cause disruptions in phosphorus uptake and metabolism [13]. In addition, phosphate ion absorption and/or even the formation of insoluble compounds occur in soils, leading to poor use of phosphorus from broadcast fertilizers [25,26]. The risk of insufficient absorption of phosphorus in the spring by young maize plants can be mitigated by early application of the so-called starter fertilization [22,23]. Some authors recommend using this method on soils low in P and/or those with factors that hinder its uptake [27]. However, as research indicates, even in conditions where soil is rich in phosphorus, band application positively affects the grain yield of cereal crops [28]. Some authors also point out that on alkaline soils, the effect of ammonium phosphates on the maize dry matter and the use of P from fertilizers depend on their chemical composition, i.e., the N:P ratio [29]. Thus, a fertilizer containing diammonium phosphate (DAP) with additional sulfur (S) seems interesting and worth investigating. The results of this study suggest that early season growth of maize in some areas of the USA may benefit from S fertilizer application [30]. Most often, the assessment of the effect of ammonium phosphates on nitrogen management is carried out in the phase of physiological maturity of the plant. However, this assessment is an ex-post analysis and allows only for a determination of the sources of the component necessary during the period of growth and pouring of seeds/kernels.

A standard NUE assessment uses partial factor productivity (PNF), agronomic efficiency (AE) and apparent N recovery efficiency (RE) indices [31]. In the author’s own study, a relatively new index was also tested, called the nitrogen gap (N_gap_), in an attempt to determine its viability in the NUE assessment in specific field experiments [32]. The value of the N_gap_ indicates the pool of nitrogen not used in crop formation, resulting either from external (e.g., dry soil) or internal (e.g., imbalance of components) factors. After flowering, the role of soil nitrogen in grain yield formation diminishes, and the importance of N remobilized from green parts increases [5,33]. Therefore, in order to fully understand the impact of phosphorus fertilization on the uptake and use of N from fertilizers, it is also necessary to take into account the growth stages preceding the maturation of plants. The character of nitrogen management can be assessed by using indicators such as nitrogen remobilization efficiency (NRE) and contribution of remobilized N to grain N content (CNR) [34]. In this research, a hypothesis was formulated that the above-mentioned indicators remain in close relation with the grain yield and nitrogen accumulation in key points of maize growth, i.e., in the sixth–seventh leaf stage and at the beginning of flowering. Furthermore, the research hypothesis states that band application of DAP with added S improves NUE and prompts the reduction in the total N rate. This assumption was based on the following facts: low mobility of P in acidic soil, the specific demand of maize for P in the initial phase of growth and the stimulating effect of S on the metabolism of N compounds [22].

The objectives of the present study were: (i) to determine the maize yield-forming reaction to increasing rates of nitrogen fertilization in comparison to various rates of band application of NP(S); (ii) to evaluate the effect of NP(S) fertilization on the nutritional condition of maize at the seventh leaf stage, beginning of flowering and technological maturity; (iii) to establish the correlation between the level of maize yield, NUE indices and nutrient content with respect to the interaction of N and P(S) fertilizations.

## 2. Results

### 2.1. Grain Yield and Crop Residue Biomass

The average grain yield (GY) in both years (2019–2020) was at a similar level of 8.45 and 8.00 t ha^−1^, respectively. The difference was 5.6%. The higher grain yield in the first year resulted from better weather and soil conditions at the start of the growing season—a damp spring and a high concentration of mineral nitrogen in soil. Unfortunately, in 2019, during intensive maize growth, kernel silking and at the beginning of kernel pouring, in June and July, weather conditions were unfavorable for the formation of the basic elements of the crop structure (Appendix A, Appendix A). Later precipitation only increased the dry matter of the vegetative parts, whereas in 2020, summer precipitation did not limit the silking and pouring of kernels. Consequently, the yield of post-harvest residue in 2019 (10.2 t ha^−1^) was considerably higher (by 34.2%) than in 2020 (*F*_1,78_ = 44.9; *p* < 0.001).

In the given years, no significant statistical differences between the fertilization treatments were confirmed. However, in each year of the study, a clear and unequivocal trend of increased GY after NP(S) band application was observed (Appendix A, Appendix A). A particularly positive impact of this type of N application was confirmed at the lowest level of nitrogen fertilization (N1). At that level, the greatest GY was obtained after an application of the highest rate of NPS (N1P3 treatment). In 2019, the difference in relation to the absolute control (N0P0) was 22.5%, and in 2020, 24.3%. At N2, high increases in GY were also observed after the application of the highest NP(S) rate (N2P3). For N3, the differences in the years were transparent. In 2019, higher rates of ammonium phosphate lowered GY. In 2020, the highest increase in GY was obtained for the N3P3 treatment. In relation to the control, the difference was 31.8%. For comparison reasons, the maximum growth of GY in the treatment without NPS, but with the highest rate of total N, was only 16.4%.

The analysis of variance showed a significant impact of the fertilizing factor on the mean values of GY for the two years (Figure 1a). Significant differences were recorded between the control (N0P0) and N2P3 and N3P1. Regardless of the N rates (without taking N0P0 into consideration), an average influence of the ammonium phosphate and ammonium sulphate mixture on GY was as follows: P0 (7.82) < P1 (8.10) < P2 (8.54) < P3 (8.81 t ha^−1^). Meanwhile, the impact of total N rates on GY was as follows: N1 (8.00) < N2 (8.33) < N3 (8.62 t ha^−1^). In the above series, GY growth as a result of the maximum NP(S) application rate was 12.7%, and the total nitrogen rate was 7.8%.

The studied factor did not differentiate the dry matter of post-harvest residue—straw yield (SDM)—in any significant way. However, fertilization clearly increased the dry matter of the vegetative parts. In both years, AP application had a positive effect on SDM, especially in N1 and N2 treatments. A difference between the years was registered for the N3 treatment. In the first year (2019), the higher rate of AP did not stimulate SDM at the same level as in 2020 (Appendix A).

On average, over the two years of study, the highest increase in SDM was obtained for N1P3, N2P2 and N2P3 treatments (Figure 1b). The difference in SDM between the nitrogen control (N0 = 7.23 t ha^−1^) and the highest rate of total nitrogen (N3 = 9.00 t ha^−1^) was approximately 24.5%, while the difference between the phosphorus control (P0 = 7.93 t ha^−1^) and P3 treatment (9.22 t ha^−1^) was approximately 16.3%. Despite this, these differences were not significant.

The biological maize yield (the sum of GY and SDM) also increased as the rates of N and P became higher, regardless of the year (Appendix A). In 2019, the highest biological yield was obtained under the N1P3 treatment, and in 2020, under N3P3.

The studied factor did not exert any major impact on the harvest index (HI). As opposed to GY and SDM, no clear trend was observed in terms of the general effect of fertilization. The parameter for N0P0 was 46.9% and 54.6%, depending on the year, whereas the maize fertilized with N and P showed HI values ranging from 41.0% (N2P2) to 49.5% (N3P2) in the first year, and from 49.7% (N2P2) to 54.7% (N3P0) in the following year.

### 2.2. Nitrogen Concentration and Accumulation at Maturity

The nitrogen content in maize significantly depended on the growing season, fertilization treatment and the studied part of the plant (Table 1). In the first year (2019), the N content in grain and in straw was higher than in 2020. At the same time, the accumulation of total N was also considerably higher in 2019 (269.9 kg ha^−1^) than in 2020 (216.9 kg N ha^−1^). The nitrogen harvest index (NHI) was, in turn, lower in 2019 (55.2%) than in 2020 (60.9%).

The studied factor notably influenced the content of N in grain (Ng) and in crop residues (Ns) in each year (Appendix A). On average, for the two years of study, the greatest content of N in grain was registered for the treatment with the maximum fertilization rate of N and P (N3P3). The use of band application of DAP caused a particular increase in Ng at the level of N1. A similar interrelation was also obtained for Ns (Table 2). As a result of the fertilization impact on N concentration in grain and in crop residues, as well as on dry matter of the earlier mentioned maize parts, real differences in the accumulation of total N (TN) were recorded. A significant increase in TN in comparison to the control (N0P0) was registered in the following treatments: N1P3, N2P2, N2P3, N3P2 and N3P3. Depending on the treatment, the difference was 51.0–56.8%.

The average impact of P(S) rates on TN over the two years was as follows: P0 (228.2) < P1 (237.2) < P2 (257.2) < P3 (274.3 kg ha^−1^). For comparison reasons, the effect of the N rates was as follows: N1 (8.00) < N2 (8.33) < N3 (8.62 t ha^−1^). The maximum difference for NPS rates was 20.2%, and for the full rate of N, it was 7.8%. The NHI values were not significantly differentiated by the application of NPS fertilization. The study did not prove any notable influence of ”year × NP(S) treatments” interaction on the content or accumulation of nitrogen.

### 2.3. Nitrogen Use Efficiency Indices

The values for nitrogen fertilization efficiency indices are shown in Table 2. It is clear that the NP(S) treatments had a significant effect on indices such as PFP, PNB, RE, IE and N_gap_. For comparison purposes, the growing season considerably influenced such indices as PNB, IE and N_gap_. The studies did not confirm any great impact of ”year × NP(S) treatments” on the values of the above-mentioned indices, however. In relation to PFP and N_gap_, it was the total N fertilization that played the vital role. Higher N rates lowered PFP, yet increased N_gap_. NP(S) fertilization brought the values of PFP up, regardless of the N fertilization levels (except for the N3P3 treatment). In relation to N_gap_, a reverse interdependence was obtained. Along with the P increase, the N_gap_ value became lower.

Negative N_gap_ values for N1 prove that N deficiency ensures a maximum yield and, at the same time, potential N soil mining, whereas for N3 and N2, the positive index values confirm the surplus of N that was not transformed into GY. The agronomic efficiency of N (AE) rose as the P rate increased. The highest AE value was obtained for the N1P3 treatment. Moreover, PNB and RE values were the highest for N1P3. The lowest values of both indices were obtained for N3, without a concomitant use of NPS. At the same time, the highest IE value was recorded in the control, while the lowest values were obtained for N2P2 and N3P3 treatments. Out of all the N and P fertilization treatments, the best use of TN was obtained for N1P0 and N1P3. Physiological N efficiency (PE) varied significantly among treatments. It is, however, worth mentioning that the lowest values were registered for the treatments without phosphorus (N1P0 and N2P0). Principal component analysis (PCA) was used to determine the relationships between the GY and NUE indices. The correlation matrix can be found in the Appendix A (Appendix A). The results of the PCA procedure were visualized in biplots (Figure 2). In addition, the interdependencies of the properties were analyzed with Pearson correlation coefficients (Appendix A).

Based on the PCA analysis, three main components, representing the GY, SDM, N content and NUE indices, accounted for 89.9% of the total variance. The first principal component (PC1) explained 39.4% of the total variability, and the next two components (PC2 and PC3), respectively, 34.6% and 18.8% of the total variance (Appendix A). As the two first principal components dominated, the results of the PCA analysis are presented on the PC1–PC2 biplot. PC1 consisted of variables related to the SDM, TN, RE and IE, as well as the GY and NHI. The loading exerted by PFP, N_gap_ and PNB influenced PC2. As shown in Figure 2, the GY was significantly related to the TN, IE, NHI and SDM. However, the GY showed negative relationships with the IE and NHI. This resulted from the fact that, in 2019, maize developed a greater biomass of crop residues than in 2020, thus lowering the share of N accumulated in grain (NHI) and N utilization efficiency (IE). The second group of parameters consists of such indices as RE, AE, PE, PNB, PFP and N_gap_. A particularly strong correlation was noted between PFP and N_gap_ parameters. The year and fertilization treatments modified the values of the investigated parameters, as demonstrated by both PCA biplots. On the PC1–PC2 biplot axes, most of the treatments were grouped closest to the Tukey median (in the bagplot) or in the bagplot cover region. Only two treatments (N3P0 in 2020 and N1P3 in 2019) were separated by a significant distance from the Tukey median. At the same time, both variants were on the opposite side of the axis representing such indices as SDM, RE and NHI. On the axis representing NUE indices on the opposite sides of the median were variants from the N3 group and variants from the N1 group but with phosphorus at the rates of P2 and P3, while on the axis representing GY, on its opposite side, treatments such as N1P0, N1P1 (2020) and N2P2, N3P3 (2019) were placed. Close to the axis but near the Tukey median, treatments such as N2P2 and N3P3 from 2020 were found.

### 2.4. Nitrogen Status of Maize

The nutritional assessment was carried out in the seventh leaf stage and at the beginning of flowering. The growing season had a significant impact on the maize dry matter in the first term, as well as on its N content (Table 3). No significant interaction was found for ”year × fertilization treatments”. Greater dry matter (DM) was obtained in the first year, however, with a slightly lower N content. As a result, N accumulation in plants in 2019 was higher than in 2020.

In general, NP(S) application improved maize nutritional status in terms of N content. However, these changes were not statistically significant. The differences in comparison to the control were particularly visible at N1, while for N3, the differences in N content were inconclusive (Appendix A).

As opposed to the first term, at the beginning of flowering, fertilization significantly differentiated N accumulation in plants (Figure 3). Both N and P fertilization increased the content of N in plants. For N1, along with the NP(S) rate increase, N accumulation rose by 13.3%. For N2, the maximum accumulation of N was recorded for treatments with P1 and P2 rates, while for N3, the highest N accumulation was registered after an application of the highest NPS rate, i.e., P3. The differences between N accumulation for P0 and other treatments were approx. 12–13%.

### 2.5. Dry Matter and N Remobilization Indices

The effect of fertilization on the average values of indices describing the management of dry matter and N after flowering is shown in Table 4. The analysis of the growing season shows that the DMI and NI indices were higher in 2019 than in 2020. At the same time, a reverse interdependence was confirmed in relation to other indices. The values of DMR, DMRE and CDMR indices were negative in the first year of the study. In 2020, the values of the indices were positive. Such correlation resulted from a high growth of vegetative maize parts after flowering. In contrast to DM, the NI and NR indices were positive regardless of the year. NRE and CNR index values also confirm that nitrogen accumulated more in the green plant parts than in grain. The fertilization factor, however, had no significant impact on their values, nor did it exert any real influence in either year of the study. Nevertheless, some real trends emerged worth describing in terms of plants’ reaction to the growing season and the NP(S) fertilization (Appendix A). In this reference, as a result of the NP(S) application, the value index of NRE increased for N1 but only continued to grow up to the rate of P2. At the higher levels of nitrogen fertilization, the highest values were obtained in the treatments with a lower rate of P (N2P1) or without band application (N3P0), whereas the values of the CNR index were the highest for N1P0, N2P1, N3P0. The lowest CNR values were obtained in the N1P3, N2P3 and N3P1 treatments but without significant differences among treatments.

In order to determine the interdependencies of the studied factors with the grain yield or the dry matter of the vegetative parts, a PCA analysis was carried out. Consequently, three factors were obtained, which, in combination, explained 94.6% of the total variance (Appendix A). However, the first two, PC1 and PC2, explained 62.3% and 20.6% of the variance, which is equal to 82.9% of the total. Therefore, PC1 and PC2 were used to explain the interrelation between the analyzed parameters and the impact of the individual fertilization treatments on the axes representing particular variances (Figure 4). The indices of N nutrient management (CNR, NRE) were significantly, in a positive way, correlated with the DMRE and CDMR indices. The indices correlated more with GY than with DMRE and CDMR. The last two indices, however, negatively correlated with the content of N in plants at the flowering growth stage. That particular parameter, in turn, positively correlated with the dry matter of post-harvest residue and, interestingly, with the plants’ dry matter at the seventh leaf stage. The values of correlation coefficients between the various features are included in the materials (Appendix A).

The PCA biplot indicates the dominant impact of the seasonal factor on the axes arrangement of the interrelations mentioned above (Figure 4). As the analysis shows, most treatments were grouped near the Tukey median. Along the GY axis, quite a distance from the median, the following treatments were placed: control (N0P0) in 2019 and 2020, and the N1P0 treatment in 2020. On the opposite side of the median, the points representing different treatments were more scattered on both sides of the GY axis and, at the same time, the treatments creating clusters were more determined by the growing season than the fertilizer rate. On the left-hand side of the GY axis, therefore, for the high GY values, the treatments representing high fertilizer rates in 2020 were present (N2P2, N3P2, N3P3). For the lower GY values, on the right-hand side of the axis, various treatments from 2019 appeared, such as N2P1, N2P2 and N3P3. This result emphasizes the role of the growing season in the modification of the interdependencies among the treatments.

## 3. Discussion

The study reveals a lack of a significant impact of the growing season on grain yield. Maize requires between 500 to 800 mm of water depending on the environment [35]. In both growing seasons, the precipitation was approximately 250–300 mm. Taking into consideration the low water retention of the sandy soil and its water demand of 500 mm, it can be assumed that in the years of the study, water deficiency reached about 200–250 mm, and a potential grain yield loss could have reached 4–5 t ha^−1^ [36]. The stages of maize susceptible to water stress are the vegetative and reproductive stages, where yield loss may be as high as 18.6–26.2% and 41.6–46.6%, respectively [37]. The least favorable weather conditions during flowering were recorded in 2020. They were, however, compensated by heavy precipitation during the grain filling stage, whereas in 2019, the drought in June and early July occurred during the most intensive period for the plant. The negative effect of the drought on GY at that stage is manifested mainly in the reduction in plant height, leaf size and delay in leaf tip emergence [38]. Nevertheless, the drought may have also had a negative impact on the tasseling growth stage and successful pollination. As a result, GY was slightly higher in 2019 than in 2020.

A considerable difference was, however, recorded in the dry matter of post-harvest residues, as the vegetative growth was stimulated by heavy precipitation in 2019. Additionally, a high content of N_min_ in the soil in 2019 largely stimulated the development of vegetative biomass. Bearing in mind the variable weather and soil conditions, the influence of N and P(S) fertilization on the maize yield was quite similar over the years of the study; no statistically significant interaction was recorded. Nonetheless, at a higher level of nitrogen fertilization (N3 = 180 kg ha^−1^ of N), the degree of plant reaction to the band application of NPS depended on the year. In 2019, the yield-forming impact of the factor on GY and SDM was lower than in 2020, which resulted from a variable N_min_ content in the soil. The optimal N rate in maize cultivation depends on many factors. The most important ones include: soil type, the course of weather, water availability, N_min_ content in soil, as well as the time and method of nitrogen application [12,39,40]. According to the literature, N fertilization greatly increases GY, most often within the rate range of up to 120–220 kg ha^−1^ N [41,42,43,44]. Research carried out in northern China showed that the economically optimal N rate may be considerably higher and depend on the soil type [45]. According to the authors, it was 265 kg ha^−1^ of N in black soil, while in aeolian sandy soil, it was 186 kg N ha^−1^. Ahmad et al. [46] point out that N applied to the soil in excessive amounts disturbs plant maturation and diminishes GY. In our own studies, the influence of the N rate on GY significantly depended on the ammonium phosphate fertilization level. On average, for the 2 years, the highest GY increase in comparison to the control was obtained for N2P3 (120 kg ha^−1^ of N) and N3P2 (180 kg ha^−1^ of N). It should, however, be emphasized that only slightly lower yield in comparison to the above-mentioned treatments was obtained for N1P3 (60 kg ha^−1^ of N). The difference was only 1.5–1.6%. Concurrently, the result indicates the dominant yield-forming impact of localized placement of NPS in comparison to the broadcast fertilization of N. In the present study, the optimal P rate depended on the level of N fertilization. For N1, the ideal rate was 26.2 kg ha^−1^ of P. For comparison purposes, during the broadcast application of P, the optimal rate in maize cultivation may reach even up to 100 kg ha^−1^ of P [47]. However, it should be stressed that the impact assessment of a single element P on maize in a ternary fertilizer mixture (N + P + S) is very difficult. Moreover, the fertilizer also included zinc (Zn). Both S and Zn have a positive effect on the uptake and metabolism of N in early growth, and consequently, on maize yielding [30,48]. The study hypothesis states, however, that the main components modifying the level of maize yielding and NUE indices are N and P. The hypothesis was based on the proven role of NH_4_^+^ and PO_4_^−3^ ions in root formation, N uptake and, as a result, maize yield. The plants react much better to N in the form of di-ammonium phosphate (DAP) than to N with ammonium nitrate and urea applied, either before sowing, as fertilizer placement at sowing or at the fifth/sixth leaf growth stage [17,20,22]. Additionally, according to Weiß et al. [23], the starter fertilization DAP increases GY to a greater extent than a simultaneous application of triple superphosphate and ammonium nitrate. The result clearly indicates a synergistic effect of NH_4_^+^ and PO_4_^−3^ ions. It is physically impossible to isolate the N or P effects in a binary fertilizer. The synergistic impact of N and P applied as DAP results from both specific transformations of NH_4_^+^ and PO_4_^−3^ ions in soil, as well as their influence on root morphology [20]. Both ions feature low effective diffusion coefficients in soils [49]. The NH_4_^+^ ions readily bind to negative charges on the surface of clay minerals and become fixed [50], while PO_4_^−3^ ions are readily fixed by adsorption to aluminum and other metal hydroxides or are precipitated depending on the pH as Fe-, Al- and Ca-phosphates [51]. According to Bordoli and Mallarino [27], P increased GY only in very low or low soil testing, and there was no response to P on any site. In our own study, despite high P concentration in the soil, P application may have had a positive impact on the maize yield, as the soil pH was acidic (5.2–5.3), and chemical sorption of P could have taken place [52]. Unlike broadcast fertilization, with or without soil incorporation, banding application reduces the surface area of contact with soil, thereby reducing PO_4_^−3^ immobilization by fixation to various cations. Moreover, both ions stimulate the initiation and elongation of lateral roots on the part of the root system that is within or close to their respective nutrient depots, which is caused by the accumulation of the plant hormone auxin [53,54]. N and P also have a positive effect on root growth in soil zones distant from the nutrient patch [16]. In order to obtain a maximum maize yield, an early supply of adequate amounts of phosphorus to the plants is crucial [13]. Placing a NP mineral fertilizer near the maize seeds leads to a higher plant-available P concentration in the soil, greater uptake by plants and a higher unit production [15,22].

A considerable difference was, however, recorded in the dry matter of post-harvest residues, as the vegetative growth was stimulated by heavy precipitation in 2019. Additionally, a high content of N_min_ in the soil in 2019 largely stimulated the development of vegetative biomass. Bearing in mind the variable weather and soil conditions, the influence of N and P(S) fertilization on the maize yield was quite similar over the years of the study; no statistically significant interaction was recorded. Nonetheless, at a higher level of nitrogen fertilization (N3 = 180 kg ha^−1^ of N), the degree of plant reaction to the band application of NPS depended on the year. In 2019, the yield-forming impact of the factor on GY and SDM was lower than in 2020, which resulted from a variable N_min_ content in the soil. The optimal N rate in maize cultivation depends on many factors. The most important ones include: soil type, the course of weather, water availability, N_min_ content in soil, as well as the time and method of nitrogen application [12,39,40]. According to the literature, N fertilization greatly increases GY, most often within the rate range of up to 120–220 kg ha^−1^ N [41,42,43,44]. Research carried out in northern China showed that the economically optimal N rate may be considerably higher and depend on the soil type [45]. According to the authors, it was 265 kg ha^−1^ of N in black soil, while in aeolian sandy soil, it was 186 kg N ha^−1^. Ahmad et al. [46] point out that N applied to the soil in excessive amounts disturbs plant maturation and diminishes GY. In our own studies, the influence of the N rate on GY significantly depended on the ammonium phosphate fertilization level. On average, for the 2 years, the highest GY increase in comparison to the control was obtained for N2P3 (120 kg ha^−1^ of N) and N3P2 (180 kg ha^−1^ of N). It should, however, be emphasized that only slightly lower yield in comparison to the above-mentioned treatments was obtained for N1P3 (60 kg ha^−1^ of N). The difference was only 1.5–1.6%. Concurrently, the result indicates the dominant yield-forming impact of localized placement of NPS in comparison to the broadcast fertilization of N. In the present study, the optimal P rate depended on the level of N fertilization. For N1, the ideal rate was 26.2 kg ha^−1^ of P. For comparison purposes, during the broadcast application of P, the optimal rate in maize cultivation may reach even up to 100 kg ha^−1^ of P [47]. However, it should be stressed that the impact assessment of a single element P on maize in a ternary fertilizer mixture (N + P + S) is very difficult. Moreover, the fertilizer also included zinc (Zn). Both S and Zn have a positive effect on the uptake and metabolism of N in early growth, and consequently, on maize yielding [30,48]. The study hypothesis states, however, that the main components modifying the level of maize yielding and NUE indices are N and P. The hypothesis was based on the proven role of NH_4_^+^ and PO_4_^−3^ ions in root formation, N uptake and, as a result, maize yield. The plants react much better to N in the form of di-ammonium phosphate (DAP) than to N with ammonium nitrate and urea applied, either before sowing, as fertilizer placement at sowing or at the fifth/sixth leaf growth stage [17,20,22]. Additionally, according to Weiß et al. [23], the starter fertilization DAP increases GY to a greater extent than a simultaneous application of triple superphosphate and ammonium nitrate. The result clearly indicates a synergistic effect of NH_4_^+^ and PO_4_^−3^ ions. It is physically impossible to isolate the N or P effects in a binary fertilizer. The synergistic impact of N and P applied as DAP results from both specific transformations of NH_4_^+^ and PO_4_^−3^ ions in soil, as well as their influence on root morphology [20]. Both ions feature low effective diffusion coefficients in soils [49]. The NH_4_^+^ ions readily bind to negative charges on the surface of clay minerals and become fixed [50], while PO_4_^−3^ ions are readily fixed by adsorption to aluminum and other metal hydroxides or are precipitated depending on the pH as Fe-, Al- and Ca-phosphates [51]. According to Bordoli and Mallarino [27], P increased GY only in very low or low soil testing, and there was no response to P on any site. In our own study, despite high P concentration in the soil, P application may have had a positive impact on the maize yield, as the soil pH was acidic (5.2–5.3), and chemical sorption of P could have taken place [52]. Unlike broadcast fertilization, with or without soil incorporation, banding application reduces the surface area of contact with soil, thereby reducing PO_4_^−3^ immobilization by fixation to various cations. Moreover, both ions stimulate the initiation and elongation of lateral roots on the part of the root system that is within or close to their respective nutrient depots, which is caused by the accumulation of the plant hormone auxin [53,54]. N and P also have a positive effect on root growth in soil zones distant from the nutrient patch [16]. In order to obtain a maximum maize yield, an early supply of adequate amounts of phosphorus to the plants is crucial [13]. Placing a NP mineral fertilizer near the maize seeds leads to a higher plant-available P concentration in the soil, greater uptake by plants and a higher unit production [15,22].

The content of nitrogen in the grain and in the post-harvest residue was higher in 2019 than in 2020. The result can be directly associated with the difference of N_min_ concentration in the soil. Fertilization treatments also exerted a great impact on the N content in grain. However, the experiment did not confirm any significant interrelation between the two factors. The average N content in grain was 17.9 and 16.3 g kg^−1^ of DM, depending on the year. According to Tenorio et al. [55], the grain of maize cultivated in the northcentral region of the USA ranged from 7.6 to 16.6 g kg^−1^. Therefore, the N concentration obtained can be considered as being in the optimal range for maize grain. In addition, this level of Ng indicates a high efficiency of the soil (even on the control) as well as fertilization treatments. The high efficiency of N accumulation in grain was assured by high concentration of K and Mg in the soil [56]. Along with the N and P(S) rates, the N concentration increased, reaching the maximum value for the N3P3 treatment. The highest accumulation of TN (276.0 kg ha^−1^ of N) was also obtained for this treatment. It resulted from the stimulation of the highest NPS rates, N uptake and dry matter accumulation as a consequence of N absorption of solar radiation during plant maturation [57]. As Figure 3 shows, GY was positively correlated with TN. The regression equation for this correlation is as follows:GY = 4.7591 + 0.0142 × TN; R^2^ = 0.72; *p* < 0.001; *n* = 26(1)

In order to assess the efficiency of nitrogen fertilization (NUE), several standard indices were used: partial factor productivity (PFP), agronomic efficiency (AE), partial N balance (PNB), apparent N recovery efficiency (RE), internal N utilization efficiency (IE) and physiological N efficiency (PE). Additionally, a relatively new index was chosen—nitrogen gap (N_gap_)—indicating potential yield losses as a result of imbalanced fertilization [32]. Among them, PFP, RE, IE and N_gap_ were considerably differentiated by fertilization factors. The values of the first three indices decreased as the N rate became lower. It is a general rule that the indices are higher with a low N rate [58]. PFP is a simple production efficiency expression, calculated in units of crop yield per unit of N applied. The advantage of PFP over others results from its sensitivity both to the course of weather and experimental factors. As the literature shows, the PFP values in maize cultivation for grain stay within the range of 6.1 and 114.9 kg kg^−1^ [14,44,59,60,61]. Thus, the PFP are within the above-mentioned range, yet they remain quite high. The result confirms the high productivity of the applied nitrogen. The experiment proved that PFP increased as the amount of NPS incorporated into the soil became higher and/or as the N share in the starter rate was greater. The differences were not significant but nevertheless clear for N1 and N2, while for N3, the differences were the smallest, clearly indicating the productivity decrease in high NPS rates when the maximum N rates were used. The other indices closely associated with the N rate—PNB and RE—point to the same trend. The expression of plant N content per unit of fertilizer N applied (PNB) indicates that, for N1, the maize effectively took advantage of the soil resources (the values were higher than 1). For N3, in turn, excessive N fertilization was recorded, expressed through values lower than 1 [62]. Regardless of the fertilization level, N exhibited a positive impact of NPS on the PNB values. With regard to RE, the index values for maize are, on average, 24.3–58.2% N [6,45,61]. Dhakal et al. [44] reported that the mean RE was 70% at the lowest N rate (60 kg ha^−1^ of N) compared to 50% at the highest N rate (240 kg ha^−1^ of N). The index is often used, as it measures the accumulation of the component in soil and/or the potential N losses to the environment [7]. In the present study, RE values were approximately 76.4% in 2019 and 60.8% in 2020. On average, for N1, N2 and N3, the index was 94.3, 63.7 and 47.7%. The use of NPS increased RE at every treatment level (and, at the same time, lowered potential losses of N to the environment), especially for the N1 treatment. For N1P3, the average value was >100%. The result can also be explained through the soil mining of N. A full confirmation of the hypothesis comes from the N_gap_ index. At the level of N1, negative values of N_gap_ were obtained and decreased along with the NPS rate. For N2 and N3, the positive values indicate the excess of N, which is the nitrogen not fully used by the plant. In the experiments, the AE index was also used to evaluate NUE. It reflects more closely the direct production impact of an applied fertilizer. Despite the fact that no statistically significant differences were obtained, a strong and clear trend of improving the index through the band application of NPS was noted. The trend was particularly transparent for N1 and then for N2. Typical AE levels of N for maize range from 1.9 to 29.0 kg grain kg^−1^ N [6,44,59,61,63]. Unlike this index, IE differentiated substantially between N and P(S) treatments. On average, over the two years of the study, lower values were recorded for N2P2 and N3P3 than for the control or for N1P0. The result indicates that 1 kg of uptaken nitrogen was more effective in terms of yield forming when NPS was being simultaneously applied. In summary, we can conclude that a shortage of P during the early growth stages results in a GY decrease, in turn negatively affecting NUE indices. The advanced procedure of NUE indices evaluation relies on the degree of their sensitivity to indicators of maize GY and N status at harvest. The PCA analysis revealed a number of interesting interrelations among the studied features. GY was positively correlated with TN, NHI and the NUE—IE index. The reliance of GY on IE and IE on NHI is described through the following equations:GY = 12.004 − 0.1094 × IE; R^2^ = 0.42; *p* < 0.001; *n* = 26(2)
IE = −10.845 + 0.7819 × NHI; R^2^ = 0.55; *p* < 0.001; *n* = 26(3)

The increase in IE is associated with an increase in NHI, which in turn was associated with a higher N translocation efficiency in later stages of the grain filling period [31,60]. In our experiment, the NPS rate increase did not lead to the improvement of NHI. The plant response to this type of fertilization depended on the level of N fertilization.

The effect of applied fertilization treatments on NUE can be considered on many different levels. There are two stages that deserve special attention: (i) N uptake and accumulation in plants; (ii) remobilization of N from vegetative to generative parts. During the first stage, a phase of intense N accumulation in plants can be distinguished. It starts right after the sixth/seventh leaf growth stage and lasts until the beginning of maize flowering [64]. The optimal N content in maize at the sixth/seventh leaf growth stage ranges from 3.5 to 5.0 g kg^−1^ [65]. In our own study, the N content was therefore within the optimal range. Maize fertilized with NPS showed a higher concentration of that component than without NPS but only for the N1 and N2 treatments. The beneficial influence of NPS was particularly transparent for N1. As for N3, a positive impact of that factor was overshadowed by high rates of N. There are several possible explanations for the positive effect of banding NPS on N uptake and concentration. At the initial growth stage, this phenomenon can be explained by an increased N_min_ and P concentration in the soil solution near the roots [54]. At the same time, ammonium-induced acidification in the rhizosphere resulted in the increased solubility of sparingly soluble P compounds, such as apatite and struvite, resulting in higher P availability compared with the supply of NO_3_-N [25,26]. Rhizosphere acidification induced by banding DAP application also increases the uptake of a greatly important element, namely Zn [17]. The authors also observed that there was a higher N uptake rate per unit of maize root biomass in response to band application of DAP compared with other treatments. The rise in the concentration of N in maize may also be explained through the modification of root architecture as a consequence of a high concentration of NH_4_-N and P in the soil layer around the developing plants. Weligama et al. [21] report that band application increases total root length and root dry matter, while according to Ma et al. [17], it also positively conditions lateral root development. Ammonium only has a minor effect on root hair length or density, and an excess of phosphorus even reduces those features [66,67]. Earlier research additionally showed a beneficial effect of DAP application on maize N nutritional status at the sixth/seventh leaf stage, expressed by the plants’ dry matter and SPAD index [14]. However, the influence depended on the depth at which the fertilizer granules were placed. In terms of N content, the optimum depth was 5–10 cm. At another critical growth stage, the beginning of flowering, localized NPS fertilization increased the total accumulation of N in plants. This was, among other factors, the consequence of better plant nutrition at an earlier stage. The amount of N at the flowering stage is particularly important, as after that stage, root activity and the ability to uptake N from the soil diminish [64]. According to Pampana et al. [34], before silking, maize uptakes approximately 64–70% of nitrogen from the soil. Most of the N in cereal kernels comes from the remobilization processes. The share of that form of N in kernels may constitute 50–90% of total nitrogen in cereal kernels [68,69]. As the main goal of maize cultivation was grain, therefore, the regression curves were determined to show the dependence of GY on N accumulation at the sixth/seventh leaf stage and flowering:6/7 leaf: GY = 5.436 + 8.092 × Na1; R^2^ = 56; *p* < 0.001; *n* = 26(4)
Flowering: GY = 5.14 + 1.401 × Na2; R^2^ = 46; *p* < 0.001; *n* = 26 (5)

In terms of plant nutrition diagnostics, the first interdependence is vital, because it is the last growth stage to change that in agricultural practice.

In order to determine the impact of N management on maize yielding, several indices were used. As far as the physiology of yielding is concerned, the following factors are considered largely relevant: dry matter remobilization efficiency (DMRE); contribution of DMR assimilates to grain (CDMR); nitrogen remobilization efficiency (NRE); contribution of remobilized N to grain (CNR). Earlier research confirmed that depending on the hybrid maturity class of maize, indices’ values DMRE, CDMR, NRE and CNR can be 16.9–24.5%, 24.9–35.8%, 38.0–44.4% and 40.2–50.3%, respectively [34]. In comparison to these values, our own study obtained similar values for NRE and CNR. Nevertheless, DMRE and CDMR were negative. In 2019, the increase in the total maize biomass after anthesis (positive values of DMA) resulted mainly from the rise of the green parts of biomass. Thus, the DMR index was negative and, consequently, so were the DMRE and CDMR indices. This means that in 2019, the current photosynthesis was significantly responsible for the accumulation of DM in kernels. Such a reaction of plants is characteristic for the ”stay green” types [5]. Ray et al. [33] also obtained negative results of DM remobilization indices as a consequence of large increases in maize biomass in the post-silking period. Unlike DM management indices, the nitrogen management indices (NRE and CNR) obtained in our studies were positive. These values confirm that N accumulation in kernels mainly relies on N remobilization. At the same time, in 2019, the remobilization index was considerably lower than in 2020, which should be associated with various concentration levels of N_min_ in the soil [5]. When maize N uptake is sustained throughout the grain filling period, less N is mobilized from vegetative organs, thereby increasing leaf area duration, delaying senescence and enhancing dry matter accumulation in grains [70]. In the conducted research, the fertilization factor changed the values of NRE and CNR indices in a particular way, usually making changes only up to a certain level. At the level of N1, the highest NRE value was obtained for the P2 rate, and at the level of N2, for the P1 rate, whereas for CNR, the optimum treatment was N2P1, followed by N1P0. The directions of the index value changes clearly indicate a competition between the main plant parts to obtain nitrogen and carbon, most probably determined by the plants’ N nutritional status. This is confirmed by the lack of a positive correlation between the mentioned indices and the grain yield. As the earlier studies confirm, remobilization of N from vegetative plant parts was covered mainly by depletion of stem N at high N supply and by depletion of leaf N at low N supply [34,71]. If the main source of remobilized N was leaves, this brought about early leaf senescence and led to the decreasing accumulation of dry matter in grains from the current photosynthesis [72]. In the present studies, rising NPS rates boosted N accumulation in plants; therefore, the potential of utilizing N from leaves was high, but the real translocation of N to kernels was lower due to a higher metabolic activity of leaves well nourished in N and P(S). Our studies confirm earlier observations, where fertilization treatments without phosphorus (NK), indices of remobilization and contribution activities were higher than in treatments without nitrogen but with phosphorus (PK) [33]. At the same time, our studies indicate that N management indices do not have to have a linear correlation with NPS rates.

## 4. Materials and Methods

### 4.1. Experiment Location and Design

The experiment was carried out in 2019–2020 at Brody, Poznan University of Life Sciences Experimental Station, in Poland (52°26′18″ N 16°17′40″ E). The following factors were tested: band application of ternary fertilizer mixture (NPS) and different total N rates. Treatments of NP(S) fertilization were as follows: P_0_—control; P_1_—8.7; P_2_—17.4; and P_3_—26.2 kg ha^−1^ of P (17%, 33% and 67% of total P uptake for grain yield = 9.0 t ha^−1^). The N rates were as follows: N_0_—control; N_1_—60; N_2_—120; and N_3_—180 kg ha^−1^ of N. A nitrogen rate of 180 kg ha^−1^ corresponded to 100% of maize requirement for this nutrient in mineral fertilizers. The effect of mineral fertilization with NP(S) was compared to the absolute control (AC), without fertilization with these components. NP(S) were applied to the soil using fertilizer NPS(+Zn) in the proportion 20-20-35+0.3 (calculated for N, P_2_O_5_, SO_3_, Zn). The fertilizer is based on two main chemical compounds: di-ammonium phosphate, DAP [(NH_4_)_2_HPO_4_] and ammonium sulfate [(NH_4_)_2_SO_4_]. Therefore, on plots with localized fertilization P_1_, P_2_ and P_3_, the doses of ammonium nitrogen (NH_4_-N) were 20, 40 and 60 kg ha^−1^ of N. A detailed summary of the doses of N, P, S and Zn used in the experiment is given in Table 5. In order to simplify the notation of fertilization treatments, the NPSZn rates were recorded as P rates.

The NP(S) fertilizer was applied while sowing the seeds. The fertilizer was banded to 5 cm below and away from the seeds. In treatments without NP(S) fertilization or with low total N rates, the maize requirement for N was supplemented with ammonium nitrate (34% N)—broadcast application immediately before sowing. Potassium fertilization was carried out in early spring in the form of potassium salt K(S, Mg) in proportion 41(15, 6.5) at the rate of 66.4 kg ha^−1^ of K, regardless of the NPS treatment.

A randomized complete block design (RCBD) with four replicates was applied in the experiment. The area of an individual plot was 24.0 m^2^ (3 × 8 m). The variety *Zea mays* L. “ES Zizou” was used in the experiment (Euralis Semences, Lescar, France), which is a variety of FAO 220 earliness; type of seeds: flint-dent/flint; average early vigor and ”stay green” type. Maize was also a forecrop. Sowing was performed on 20 April 2019 and 21 April 2020 with 95,000 seeds per hectare. The row spacing was 0.75 m, while the stem spacing was 0.14 m. Plant protection included the application of herbicide (mixture of terbuthylazine, S-metolachlor and mesotrione) before the emergence of the plants.

### 4.2. Soil and Meteorological Conditions

According to the World Reference Base for Soil Resources [73] classification system, the soil in the experiment was classified as Haplic Luvisols. The topsoil was characterized by sandy loam, and the subsoil was characterized by loam texture. The soil organic matter content in the topsoil was around 1.37%. To assess the pH and plant-available nutrient contents, soil samples were collected each year in early spring (March/April), before the application of mineral fertilizers. The soil pH at a depth of 0–0.3 m was acidic in each season. According to the Polish classification [74], the contents of plant-available P in the topsoil were high, irrespective of the year. The content of K in the first year was medium but in the second, very high. The content of Mg was very high; Ca—low; micronutrients—medium; and S—medium. The total content of N_min_ (sum of NH_4_-N and NO_3_-N in soil depth 0.0–0.9 m) ranged from 33.0 to 93.6 kg ha^−1^ of N (Table 6).

The characteristics of climatic conditions were based on data from the meteorological station belonging to the Poznan University of Life Sciences Experimental Station in Brody (west Poland). The long-term (1960–2018) average yearly precipitation and temperature in the study area are about 590 mm and 8.5 °C, respectively. In 2019 and 2020, the sum of precipitation was 403 and 496, and the average temperature 12.0 and 11.9 °C. In these years, the sum of precipitation during the growing season (April–September) was 253 and 313 mm, respectively. For comparison, the sum of precipitation for the long-term period was 356 mm. Both growing seasons also differed in their rainfall distribution. In the first year, May had better conditions for maize growth than in 2020. In turn, in the second year, the weather conditions were better in June and August. However, rainfall during flowering was greater in 2019 than in 2020 (Figure 5). Mean temperature during the maize growing season was 16.5 °C in 2019 and 15.7 °C in 2020.

### 4.3. Plant Sampling and Analysis

Plant samples were collected at the following growth stages: seventh leaf (BBCH 17—coding system of growth stages, abbreviation (in German): Biologische Bundesanstalt, Bundessortenamt und CHemische Industrie; [77]); beginning of flowering (BBCH 61/62) and technological maturity (BBCH 91/92). Depending on the year (2019 and 2020), the sampling dates were as follows: 8 and 16 June, 19 and 27 July, 27 September and 6 October. In order to obtain the grain yield, maize cobs were hand harvested from an area of 5.25 m^2^ (2 rows × 3.5 m). Next, the cobs were counted and threshed using the laboratory threshing machine; the moisture of seeds was also determined. The biomass of plant vegetative parts was determined by randomly sampling 5 plants from each plot, regardless of the date of sample collection. The plant samples were dried at 60 °C and ground afterward. Nitrogen contents were determined with the Kjeldahl method using a Kjeltec Auto 1031 Analyzer (Foss Tecator AB, Hoganas, Sweden). Total nitrogen uptake (TN) was calculated by summing up the amount of nitrogen accumulated in grains (GN) and in the post-harvest residues (SN).

### 4.4. Nitrogen Use Efficiency Indices

In order to assess the effectiveness of nitrogen (N) fertilization, the following standard indices were used [62]:partial factor productivity of N (PFP) = GY/N_f_ [kg seeds kg^−1^ N_f_] (6)
agronomic efficiency of N (AE) = (GY − GY_0_)/N_f_ [kg seeds kg^−1^ N_f_] (7)
partial N balance (PNB) = TN/N_f_ [kg N kg^−1^ N_f_] (8)
apparent N recovery efficiency (RE) = (TN − TN_0_)/N_f_ [kg TN kg^−1^ N_f_] (9)
internal N utilization efficiency (IE) = GY/TN [kg seeds kg^−1^ TN] (10)
physiological N efficiency (PE) = (GY − GY_0_)/(TN − TN_0_) [kg seeds kg^−1^ TN] (11)
where GY—grain yield (kg ha^−1^); GY_0_—grain yield in treatment without N application (kg ha^−1^); N_f_—input of N in mineral fertilizers (kg ha^−1^); TN—total N accumulation in above-ground biomass of maize (kg ha^−1^); TN_0_—total N accumulation on control (kg ha^−1^).

Additionally, the efficiency of N can be determined using the nitrogen gap (N_gap_) index based on the concept proposed by Grzebisz and Łukowiak [32]. The following set of equations can be used to calculate N_gap_:maximum attainable yield (GY_max_) = cPFP ∙ N_f_ [kg ha^−1^] (12)
grain yield gap (GY_gap_) = GY_max_ − GY_a_ [kg ha^−1^] (13)
nitrogen gap (N_gap_) = GY_gap_/cPFP [kg ha^−1^] (14)
where cPFP is the third quartile (Q3) of the partial factor productivity (PFP) values measured for each plot with N fertilization (kg seeds kg^−1^ N_f_); N_f_—input of N in mineral fertilizers (kg ha^−1^); GY_a_—grain yield on each plot in a particular growing season (kg ha^−1^).

### 4.5. Dry Matter and Nitrogen Management Indices

The assessment of the impact of fertilization on the method of dry matter (DM) and N management by maize was based on the following indices [68]:dry matter increase (DMI) = DM_H_ − DM_A_ [g plant^−1^](15)
dry matter remobilization (DMR) = DM_A_ − DM_S_ [g plant^−1^] (16)
dry matter remobilization efficiency (DMRE) = (DMR/DM_A_) × 100 [%] (17)
contribution of DMR assimilates to grain (CDMR) = (DMR/DM_G_) × 100 [%] (18)
nitrogen increase (NI) = TN_H_ − TN_A_ [g plant^−1^] (19)
nitrogen remobilization (NR) = TN_A_ − N_S_ [g plant^−1^] (20)
nitrogen remobilization efficiency (NRE) = (NR/TN_A_) × 100 [%] (21)
contribution of remobilized N to grain (CNR) = (NR/N_G_) × 100 [%] (22)
where DM_H_—total above-ground biomass (dry matter) at maturity (g plant^−1^); DM_A_—total above-ground biomass at anthesis (g plant^−1^); DM_S_—dry matter of crop residues (stems, leaves, husk, corncob) at maturity (g plant^−1^); DM_G_—dry matter of grains at maturity (g plant^−1^); TN_H_—total N content of above-ground biomass at maturity (g plant^−1^); TN_A_—total N content of above-ground biomass at anthesis (g plant^−1^); N_S_—N content of crop residues (stems, leaves, husk, corncob) at maturity (g plant^−1^); N_G_—N content of grains at maturity (g plant^−1^).

### 4.6. Statistical Analysis

The effects of individual research factors (year, NP treatments) and their interactions were assessed by means of the two-way ANOVA. If the F-ratio was larger than the critical value (*p* < 0.05), the differences between the treatments were evaluated by using the HSD (Tukey’s) test (for α = 0.05). The distribution of the data was checked using the Shapiro–Wilk test and the homogeneity of variance with the Bartlett test. Standard error of the mean (SEM) was used to indicate statistical error. Principal component analysis (PCA) was applied for evaluation of the relationships between variables. The Tukey median is surrounded by a bag containing 50% of the data points. The bagplot visualizes the location, spread, correlation, skewness and tails of data. The bagplot cover contains the inliers and outside of the “fence” are outliers [78]. In addition, the relationships between traits were analyzed using Pearson’s correlation and linear regression. Statistica 13 software (TIBCO Software Inc., USA) was used for all statistical analyses [79].

## 5. Conclusions

The effect of nitrogen application on the maize yield and nitrogen management indices depended on the amount of mineral nitrogen in the soil, nitrogen doses and the method of its application. The most effective use of nitrogen by corn was ensured by the use of ammonium phosphate and ammonium sulphate during the sowing of corn seeds (band application). From the point of view of NUE indices, the optimal dose of N was 60 kg ha^−1^. With broadcast fertilization and/or a further increase in the N dose, without the simultaneous use of P and S, the values of NUE indices deteriorated, especially in the year with the highest content of N_min_ in the soil. Thus, a positive effect of the interaction of N and P(S) was confirmed in the conditions of soil rich in plant-available phosphorus.

Band application had a particularly positive effect on total N accumulation, nitrogen harvest index and internal N utilization efficiency. These parameters were closely related to the grain yield. For diagnostic purposes, the accumulation of N in the above-ground part of maize at the sixth–seventh leaf stage was important.

## Figures and Tables

**Figure 1 plants-11-01660-f001:**
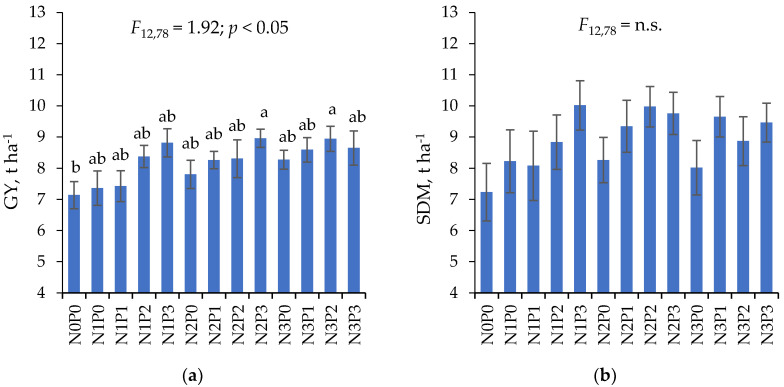
Maize grain yield, GY (**a**) and crop residues biomass, SDM (**b**) depending on fertilization treatments. Two-year average values. Means within a column followed by the same letter indicate a lack of significant difference between the fertilized treatments (HSD test). Hatched bars represent 2 × SEM ranges.

**Figure 2 plants-11-01660-f002:**
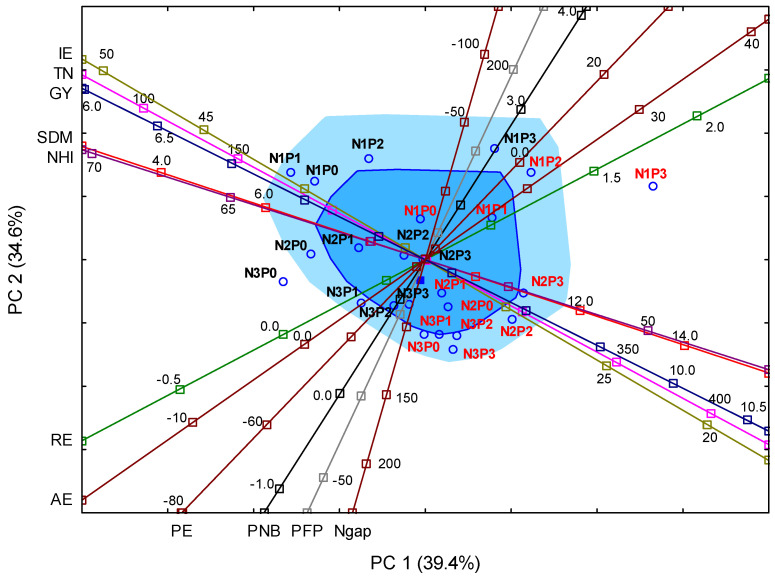
Principal component analysis (PCA) biplot of the maize yield and N use efficiency indices. The dark blue square denotes the Tukey median, the blue square is the bagplot, the light blue square is the bagplot cover. Key: GY—grain yield; SDM—crop residues (straw) biomass; TN—total N accumulation; NHI—N harvest index; PFP—partial factor productivity of N; AE—agronomic efficiency of N; PNB—partial N balance; RE—apparent N recovery efficiency; IE—internal N utilization efficiency; N_gap_—N gap. The treatments in 2019 are marked in red and the treatments in 2020 in black.

**Figure 3 plants-11-01660-f003:**
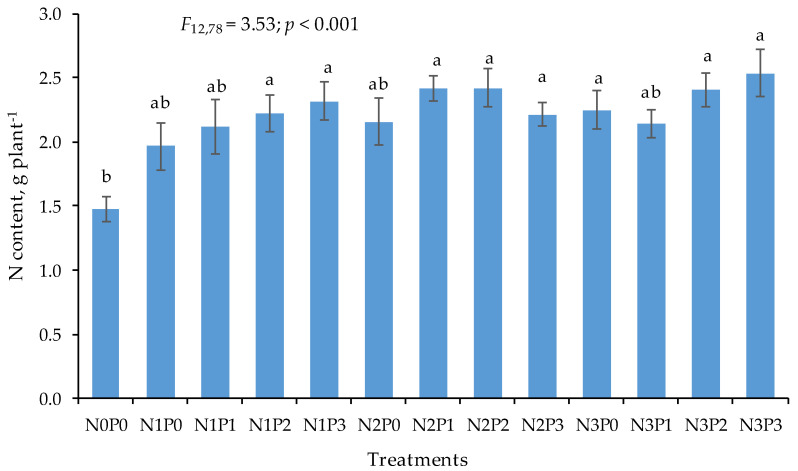
Content of nitrogen (N) in maize at the beginning of flowering depending on fertilization treatments. Two-year average values. Means within a column followed by the same letter indicate a lack of significant difference between the fertilized treatments (HSD test). Hatched bars represent 2 × SEM ranges.

**Figure 4 plants-11-01660-f004:**
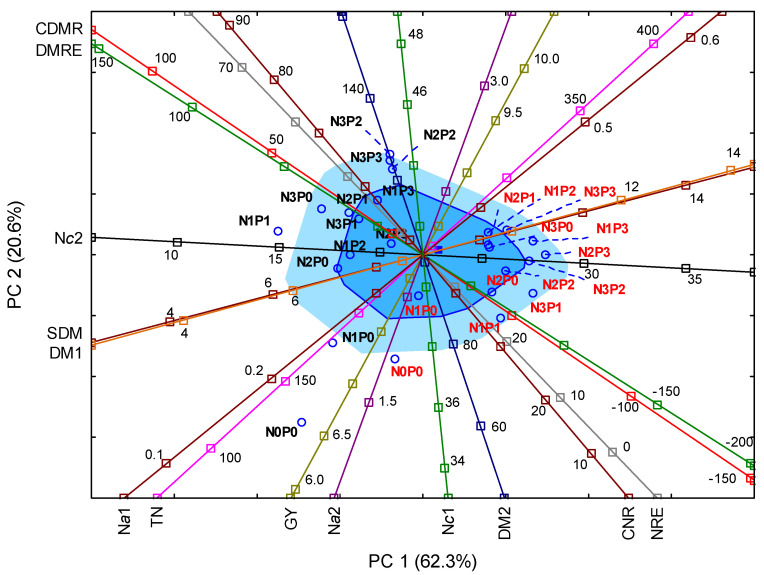
Principal component analysis (PCA) biplot of the maize yield, N content and accumulation, and N management indices. The dark blue square denotes the Tukey median, the blue square is the bagplot, the light blue square is the bagplot cover. Key: GY—grain yield; SDM—crop residues (straw) biomass; TN—total N accumulation; DM1—dry matter of plant at seventh leaf growth stage; N*c*1—N concentration at seventh leaf growth stage; N*a*1—N accumulation at seventh leaf growth stage; DM2—dry matter of plant at flowering; N*c*2—N concentration at flowering; N*a*1—N accumulation at flowering; DMRE—dry matter remobilization efficiency; CDMR—contribution of DMR assimilates to grain; NRE—N remobilization efficiency; CNR—contribution of remobilized N to grain N content. The treatments in 2019 are marked in red and the treatments in 2020 in black.

**Figure 5 plants-11-01660-f005:**
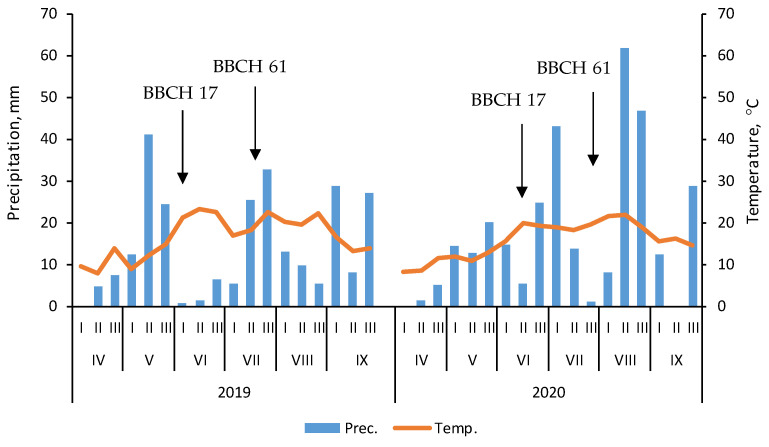
Air temperature and precipitation during 2019–2020 growing seasons of maize (in decades of the month). Key growth stages: BBCH 17—seventh leaf growth stage; BBCH 61—onset of flowering. Source: Poznan University of Life Sciences Experimental Station in Brody.

**Table 1 plants-11-01660-t001:** Nitrogen (N) concentration and accumulation in maize at maturity (harvest) depending on fertilization treatments.

Year Treatment	N in Grain g kg^−1^	N in Crop Residues g kg^−1^	N in Grain kg ha^−1^	N in Crop Residues kg ha^−1^	Total N kg ha^−1^	N Harvest Index %
Effect of the year (Y)						
2019	17.5 ± 0.3 ^a^	11.9 ± 0.3	148.2 ± 4.0 ^a^	121.7 ± 4.2 ^a^	269.9 ± 7.4 ^a^	55.2 ± 0.7 ^a^
2020	16.4 ± 0.3 ^b^	11.2 ± 0.2	131.5 ± 3.4 ^b^	85.3 ± 3.0 ^b^	216.9 ± 5.9 ^b^	60.9 ± 0.6 ^b^
Effect of the treatment (T)						
N0P0	14.3 ± 0.5 ^b^	10.3 ± 0.2	102.1 ± 6.7 ^d^	73.9 ± 9.1 ^b^	176.0 ± 15.3 ^b^	58.8 ± 1.6
N1P0	15.5 ± 0.8 ^ab^	11.1 ± 0.6	114.6 ± 9.5 ^cd^	92.1 ± 16.5 ^ab^	206.6 ± 25.6 ^ab^	55.4 ± 2.0
N1P1	16.4 ± 0.5 ^ab^	11.6 ± 0.8	122.1 ± 8.6 ^bcd^	95.6 ± 14.7 ^ab^	217.6 ± 21.8 ^ab^	56.1 ± 2.8
N1P2	16.8 ± 0.6 ^ab^	10.4 ± 0.7	141.4 ± 7.3 ^abc^	90.3 ± 7.3 ^ab^	231.6 ± 13.5 ^ab^	61.0 ± 1.4
N1P3	17.4 ± 0.4 ^ab^	12.0 ± 0.7	153.4 ± 8.3 ^ab^	121.3 ± 11.6 ^a^	274.7 ± 18.5 ^a^	55.8 ± 1.6
N2P0	16.7 ± 1.3 ^ab^	12.6 ± 0.1	130.9 ± 9.5 ^abc^	104.3 ± 9.6 ^ab^	235.2 ± 18.3 ^ab^	55.6 ± 1.6
N2P1	17.0 ± 0.7 ^ab^	10.5 ± 0.5	140.3 ± 5.4 ^abc^	99.0 ± 9.9 ^ab^	239.3 ± 14.6 ^ab^	58.6 ± 1.6
N2P2	17.7 ± 0.7 ^ab^	11.8 ± 0.6	146.9 ± 11.1^abc^	118.8 ± 9.8 ^a^	265.7 ± 18.1 ^a^	55.3 ± 2.3
N2P3	16.7 ± 0.7 ^ab^	12.2 ± 0.5	149.8 ± 5.1 ^abc^	119.8 ± 9.1 ^a^	269.6 ± 13.7 ^a^	55.6 ± 1.3
N3P0	17.6 ± 0.6 ^ab^	11.9 ± 0.9	145.9 ± 6.5 ^abc^	96.8 ± 12.6 ^ab^	242.7 ± 18.8 ^ab^	60.1 ± 2.1
N3P1	17.6 ± 1.1 ^ab^	10.8 ± 0.5	151.4 ± 8.9 ^abc^	103.1 ± 5.8 ^ab^	254.6 ± 13.8 ^a^	59.5 ± 0.9
N3P2	17.8 ± 0.3 ^ab^	13.0 ± 0.4	159.0 ± 7.4 ^ab^	115.4 ± 10.5 ^ab^	274.3 ± 13.9 ^a^	57.9 ± 2.4
N3P3	18.5 ± 0.2 ^a^	12.1 ± 0.6	160.5 ± 10.3 ^a^	115.5 ± 9.0 ^ab^	276.0 ± 15.7 ^a^	58.2 ± 2.1
ANOVA results (significance level)						
Year	*	n.s.	***	***	***	***
Treatment	*	n.s.	***	**	***	n.s.
Y × T	n.s.	n.s.	n.s.	n.s.	n.s.	n.s.

***, **, * significant at *p* < 0.001, *p* < 0.01, *p* < 0.05, respectively; n.s.—non-significant; means within a column followed by the same letter indicate a lack of significant difference between the fertilized treatments (HSD test).

**Table 2 plants-11-01660-t002:** Nitrogen use efficiency (NUE) indices depending on fertilization treatment.

Year Treatment	PFP kg kg^−1^	AE kg kg^−1^	PNB kg kg^−1^	RE kg kg^−1^	IE kg kg^−1^	PE kg kg^−1^	Ngap kg N ha^−1^
Effect of the year (Y)							
2019	86.3 ± 6.1	10.3 ± 3.4	1.51 ± 0.10 ^a^	0.81 ± 0.14	28.6 ± 0.39 ^b^	13.4 ± 1.8	53.2 ± 7.2 ^a^
2020	80.8 ± 5.3	11.3 ± 2.1	1.32 ± 0.08 ^b^	0.69 ± 0.07	35.5 ± 0.43 ^a^	18.4 ± 1.0	48.2 ± 6.8 ^b^
Effect of the treatment (T)							
N0P0	--	--	--	--	41.3 ± 1.5 ^a^	21.3 ± 22.4	*--*
N1P0	122.8 ± 9.2 ^a^	3.8 ± 13.3	1.91 ± 0.16 ^c^	0.51 ± 0.51 ^ab^	36.9 ± 1.7 ^b^	−108.6 ± 146.8	−1.4 ± 4.5 ^c^
N1P1	123.8 ± 8.6 ^a^	4.8 ± 7.8	2.03 ± 0.14 ^bc^	0.69 ± 0.33 ^ab^	35.2 ± 2.0 ^bc^	28.3 ± 7.5	−1.8 ± 4.0 ^c^
N1P2	139.6 ± 6.0 ^a^	20.6 ± 5.7	2.36 ± 0.12 ^ab^	0.93 ± 0.17 ^ab^	36.5 ± 1.1 ^ab^	20.3 ± 2.8	−9.6 ± 2.3 ^c^
N1P3	146.9 ± 7.5 ^a^	27.9 ± 10.5	2.56 ± 0.14 ^a^	1.64 ± 0.30 ^a^	32.4 ± 1.2 ^bc^	10.1 ± 6.8	−13.4 ± 3.8 ^c^
N2P0	65.0 ± 3.8 ^bc^	5.6 ± 5.8	1.09 ± 0.08 ^de^	0.49 ± 0.16 ^ab^	33.8 ± 1.8 ^bc^	−197.5 ± 131.6	55.0 ± 3.6 ^b^
N2P1	68.9 ± 2.3 ^bc^	9.4 ± 5.3	1.17 ± 0.04 ^de^	0.53 ± 0.16 ^ab^	35.0 ± 1.3 ^bc^	15.2 ± 7.6	51.0 ± 2.7 ^b^
N2P2	69.2 ± 5.1 ^bc^	9.8 ± 7.4	1.22 ± 0.09 ^de^	0.75 ± 0.18 ^ab^	31.4 ± 1.7 ^c^	−75.5 ± 91.0	50.6 ± 5.2 ^b^
N2P3	74.7 ± 2.4 ^b^	15.2 ± 4.1	1.25 ± 0.04 ^d^	0.78 ± 0.14 ^a^	33.5 ± 0.8 ^bc^	16.9 ± 3.6	45.4 ± 2.2 ^b^
N3P0	46.0 ± 1.7 ^c^	6.3 ± 3.0	0.81 ± 0.04 ^e^	0.37 ± 0.12 ^b^	34.9 ± 1.6 ^bc^	16.7 ± 5.8	111.1 ± 2.4 ^a^
N3P1	47.7 ± 2.2 ^c^	8.1 ± 3.4	0.84 ± 0.05 ^de^	0.44 ± 0.11 ^b^	33.9 ± 0.9 ^bc^	4.5 ± 11.1	108.4 ± 3.1 ^a^
N3P2	49.7 ± 2.2 ^bc^	10.0 ± 3.4	0.88 ± 0.04 ^de^	0.55 ± 0.12 ^ab^	32.9 ± 1.4 ^bc^	11.7 ± 8.0	105.4 ± 3.4 ^a^
N3P3	48.1 ± 3.0 ^c^	8.4 ± 3.9	0.89 ± 0.06 ^de^	0.56 ± 0.12 ^ab^	31.4 ± 1.1 ^c^	−10.0 ± 23.6	107.6 ± 5.1 ^a^
ANOVA results (significance level)							
Year	n.s.	n.s.	***	n.s.	***	n.s.	*
Treatment	***	n.s.	***	*	***	***	***
Y × T	n.s.	n.s.	n.s.	n.s.	n.s.	n.s.	n.s.

***, * significant at *p* < 0.001, *p* < 0.05, respectively; n.s.—non-significant; means within a column followed by the same letter indicate a lack of significant difference between the fertilized treatments (HSD test). PFP—partial factor productivity of N; AE—agronomic efficiency of N; PNB—partial N balance; RE—apparent N recovery efficiency; IE—internal N utilization efficiency; N_gap_—N gap.

**Table 3 plants-11-01660-t003:** Effect of the growing season and fertilization treatments on maize dry matter (DM) and nitrogen (N) status at the seventh leaf stage and at the beginning of flowering.

Year Treatment	Seventh Leaf Stage			Beginning of Flowering Stage		
	DM g Plant^−1^	N Content g kg DM^−1^	N Content g Plant^−1^	DM g Plant^−1^	N Content g kg DM^−1^	N Content g plant^−1^
Effect of the year (Y)						
2019	9.57 ± 0.31 ^a^	40.0 ± 0.5 ^b^	0.39 ± 0.01 ^a^	89.0 ± 2.3 ^b^	26.1 ± 0.5 ^a^	2.33 ± 0.07 ^a^
2020	7.18 ± 0.23 ^b^	42.1 ± 0.7 ^a^	0.30 ± 0.01 ^b^	114.5 ± 2.9 ^a^	18.0 ± 0.4 ^b^	2.08 ± 0.06 ^b^
ANOVA results (significance level)						
Year	***	*	***	***	***	**
Treatment	n.s.	n.s.	n.s.	n.s.	n.s.	***
Y × T	n.s.	n.s.	n.s.	n.s.	n.s.	n.s.

***, **, * significant at *p* < 0.001, *p* < 0.01, *p* < 0.05, respectively; n.s.—non-significant; means within a column followed by the same letter indicate a lack of significant difference between the fertilized treatments (HSD test).

**Table 4 plants-11-01660-t004:** Effect of the growing season and fertilization treatments on maize dry matter (DM) and nitrogen (N) management indices.

Year Treatment	Dry Matter Management Indices				Nitrogen Management Indices			
	DMA	DMR	DMRE	CDMR	NA	NR	NRE	CNR
Effect of the year (Y)								
2019	165.6 ± 7.4 ^a^	−50.4 ± 5.0 ^b^	−61.5 ± 7.0 ^b^	−40.6 ± 4.3 ^b^	1.35 ± 0.12 ^a^	0.70 ± 0.08 ^b^	26.9 ± 3.0 ^b^	39.7 ± 5.6 ^b^
2020	93.8 ± 3.9 ^b^	14.1 ± 2.9 ^a^	11.3 ± 2.4 ^a^	14.1 ± 2.9 ^a^	0.82 ± 0.06 ^b^	0.98 ± 0.05 ^a^	44.7 ± 1.7 ^a^	55.3 ± 3.0 ^a^
ANOVA results (significance level)								
Year	***	***	***	***	***	*	***	*
Treatment	n.s.	n.s.	n.s.	n.s.	n.s.	n.s.	n.s.	n.s.
Y × T	n.s.	n.s.	n.s.	n.s.	n.s.	n.s.	n.s.	n.s.

***, * significant at *p* < 0.001, *p* < 0.05, respectively; n.s.—non-significant; means within a column followed by the same letter indicate a lack of significant difference between the fertilized treatments (HSD test). DMI—dry matter increase; DMR—dry matter remobilization; NI—nitrogen increase; NR—nitrogen remobilization; DMRE—dry matter remobilization efficiency; CDMR—contribution of DMR assimilates to grain; NRE—N remobilization efficiency; CNR—contribution of remobilized N to grain N content.

**Table 5 plants-11-01660-t005:** Rates of nutrients (N, P, S and Zn) depending on the fertilization treatment.

Treatment	N (Band Appl. *) kg ha^−1^	P kg ha^−1^	S kg ha^−1^	Zn g ha^−1^
N0P0	0	0	0	0
N1P0	60 (0)	0	0	0
N1P1	60 (20)	8.7	14	300
N1P2	60 (40)	17.4	28	600
N1P3	60 (60)	26.2	42	900
N2P0	120 (0)	0	0	0
N2P1	120 (20)	8.7	14	300
N2P2	120 (40)	17.4	28	600
N2P3	120 (60)	26.2	42	900
N3P0	180 (0)	0	0	0
N3P1	180 (20)	8.7	14	300
N3P2	180 (40)	17.4	28	600
N3P3	180 (60)	26.2	42	900

* Nitrogen band application of di-ammonium phosphate (DAP) and ammonium sulfate.

**Table 6 plants-11-01660-t006:** Agrochemical properties of soil prior to experiment.

	Year/Soil Depth (m)					
	2019			2020		
	0.0–0.3	0.3–0.6	0.6–0.9	0.0–0.3	0.3–0.6	0.6–0.9
pH *	5.3	5.6	5.6	5.2	5.2	5.3
P, mg kg^−1^	208.6	132.2	43.8	166.0	62.4	22.0
K, mg kg^−1^	138.4	166.8	120.3	229.4	73.6	69.1
Mg, mg kg^−1^	104.4	120.9	105.9	150.6	129.3	107.6
Ca, mg kg^−1^	719.5	1523.3	1562.0	620.0	716.2	833.2
Cu, mg kg^−1^	1.6	1.1	0.7	4.5	1.8	1.1
Fe, mg kg^−1^	252.5	166.6	114.5	211.4	153.4	121.8
Mn, mg kg^−1^	45.1	49.6	43.2	85.9	34.5	61.1
Zn, mg kg^−1^	3.9	2.5	1.0	2.7	1.8	1.4
NH_4_-N, kg ha^−1^	10.9	4.1	9.0	7.2	0.6	0.9
NO_3_-N, kg ha^−1^	30.3	19.4	20.0	14.9	5.3	4.1
N_min_ sum, kg ha^−1^	41.1	23.5	29.0	22.1	5.9	5.0

* soil pH was determined in suspension of 1 M KCl; plant-available nutrients using the Melich 3 method [75] and mineral nitrogen using 0.01 M CaCl_2_ [76].

## Data Availability

Data sharing not applicable.

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
