# Peer review of "Band Phosphorus and Sulfur Fertilization as Drivers of Efficient Management of Nitrogen of Maize (Zea mays L.)"

_plants, 2022, doi:10.3390/plants11131660_

Round 1

Reviewer 1 Report

This manuscript deals with the effect of band fertilization at different dosages of P and N in commercial maize crops for two years. Plant growth, grain yield and several management and nitrogen use efficiency indices due to different fertilization treatments and years are analyzed as well as its relationship with some climatic variables. It is very interesting because it is necessary to reduce the environmental costs derived from the use of mineral fertilization. The topic is within the scope of the journal, although I think it would be more suitable for an agronomy journal. The manuscript is very well structured and written, the bibliography is very up-to-date, and the tables and figures are correctly prepared and are all necessary. However, some tables and figures are proposed to be reduced in size. It also needs a revision of the format in a few aspects.

The biggest weakness that I see in the article derives from the large number of measured/calculated variables. Such a number of indices and variables make the article very extensive and many acronyms appear, so that sometimes the reader gets lost in reading. But this is not a problem because the authors have not written it well, but because of the large amount of information included. Possibly all this information could have been divided into two articles.

 I recommend a major revision before being accepted for publication. The recommendation is really between minor and major revision, but I have to click the "major" tab because some formatting issues go beyond a simple minor revision.

 See attached file for particular and minor comments. It contains paragraphs highlighted in yellow and comments.

Author Response

Response to Reviewer 1 Comments

            Thank you for the many helpful suggestions to improve the manuscript. We have carefully and thoroughly revised the manuscript in response to the reviewer suggestions. In addition, all editorial shortcomings have been corrected. We uploaded Word file with the Track Changes function and pdf file without the Track Changes function. We hope that the last pdf file will make it easier to read the final version of our manuscript.

Answers to specific remarks are given below.

Point 1: Please, do not insert symbols or names between units. According to the SI of units it is preferable to type kg ha-1 of P. Please, revise the whole manuscript.

Response 1: It has been corrected in accordance to the reviewer’s suggestion. The new version of manuscript includes units according to the SI system, i.e. kg ha-1 of P and kg ha-1 of N.

Point 2: mln is not a multiple used in SI. it is preferable to use the full word "million hectares".

Response 2: The comments has been taken into consideration and changes made accordingly.

Point 3: Do the authors mean seeds or seedlings, seeds or plants?

Response 3: Thank you for your comment. It should be seedlings.

Point 4: Fig.1. As these results are included in the text above and the differences were not significant, figure 1 can be reduced and only show figures e) and f). Therefore figures a) to d) can be removed (or moved to the Supplementary material section), including in the text the global mean values and the level of statistical significance of each figure.

Response 4: The suggestion has been taken into account in the new version of manuscript. Figures with significant differences and / or for the two-year average were left in the main text. The results for each year separately (Fig. 1a-1d) were transferred to the Supplementary Material.

Point 5: Line 157-158, On average, over the two years of study, the highest increase of SDM was obtained for N1P3; N2P2 and N2P3 treatments (Figure 1b). The difference in SDM between the nitrogen control (N0 = 7.23 t ha-1) and the highest rate of total nitrogen (N3 = 9.00 t ha-1) was approximately 24.5% while the difference between the phosphorus control (P0 = 7.93 t ha-1) and P3 treatment (9.22 t ha-1) was approximately 16.3%. Despite this, these differences were not significant.

 Response 5: The suggestion has been taken into account in the new version of manuscript. It was noted in the text that the differences were not significant: “On average, over the two years of study, the highest increase of SDM was obtained for N1P3; N2P2 and N2P3 treatments (Figure 1b). The difference in SDM between the nitrogen control (N0 = 7.23 t ha-1) and the highest rate of total nitrogen (N3 = 9.00 t ha-1) was ap-proximately 24.5% while the difference between the phosphorus control (P0 = 7.93 t ha-1) and P3 treatment (9.22 t ha-1) was approximately 16.3%. Despite this, these differences were not significant.”

Point 6: Table 3. The titles of the columns coincide, which ones corresponds to 7-leaf stage and which ones to Beginning flowering?

Response 6: The error has been corrected, and in the new version of the manuscript there is an appropriate division of columns and titles.

Point 7: Table4. Since the effect of treatment was not significant for any of these variables, this table can be reduced to three rows of results, two rows for mean values per year and one row for an overall mean value of the two years and all treatments as a whole.

The rows containing ANOVA significance level does remain on the table.

Response 7: The suggestion has been taken into account in the new manuscript. The table was reduced in accordance with the reviewer's comment. The first 3 rows were left and rows containing ANOVA significance level. For other results, see Supplementary Materials.

Point 8: The same comment for Table 3 (except for the last column on the right).

Response 8: The suggestion has been taken into account in the new manuscript. The table was reduced and the statistically insignificant results were transferred to Supplementary Materials. For the last column with significant values, a new Figure 3 was created.

Point 9: Line 284-287. The lowest CNR values were obtained in the N1P3, N2P3 and N3P1 treatments, but without significant differences among treatments. But without significant differences among treatments.

Response 9: The suggestion has been taken into account in the new manuscript.

Point 10: Table 5. Please, check the units "g kg-1" They are very high levels of mineral nutrients. For instance, there are up to 1562 g kg-1 of Ca.

Response 10: Thanks for your comment. It was an editing error. It should be mg / kg of course.

Point 11: Line 699. Superscript

Response 11: The error has been corrected.

Reviewer 2 Report

Some editing of English usage would make the manuscript easier to follow and understand. The Introduction and Conclusion sections, in particular, are difficult because much of the material is not broken into paragraphs but in included in a couple of very long paragraphs that are difficult to navigate. I think the general content and completeness of the analysis is very good.

To make it easier to follow, maybe a table of describing the nutrient levels in the different fertilizer mixes could be included (probably in Methods).

Author Response

Response to Reviewer 2 Comments

            Thank you for the many helpful suggestions to improve the manuscript. We have carefully and thoroughly revised the manuscript in response to the reviewer suggestions. In addition, all editorial shortcomings have been corrected. We uploaded Word file with the Track Changes function and pdf file without the Track Changes function. We hope that the last pdf file will make it easier to read the final version of our manuscript.

Answers to specific remarks are given below.

Point 1: Some editing of English usage would make the manuscript easier to follow and understand. The Introduction and Conclusion sections, in particular, are difficult because much of the material is not broken into paragraphs but in included in a couple of very long paragraphs that are difficult to navigate. I think the general content and completeness of the analysis is very good.

Response 1: This suggestion have been considered in the new version of the paper. In the new version of the manuscript the Introduction and Conclusion sections are divided into sections on specific topics, research problems and / or results.

Point 2: To make it easier to follow, maybe a table of describing the nutrient levels in the different fertilizer mixes could be included (probably in Methods).

Response 2: Thank you for your valuable comment. The appropriate table (number 5) has been included in the new version of the manuscript.

Round 2

Reviewer 1 Report

The authors have submitted a revised and improved version of the previous one. They have taken into account the suggestions of the reviewers. Therefore, in my opinion, it is acceptable to be published.